# Perspectives on factors influencing transmission of COVID-19 in Zambia: a qualitative study of health workers and community members

Cephas Sialubanje [ID] ,[1] Doreen C Sitali,[2] Nawa Mukumbuta,[1] Libonda Liyali,[3] Phyllis Ingutu Sumbwa,[4] Harvey Kakoma Kamboyi,[3] Mary Ng'andu,[1] Fastone Matthew Goma[3]

[1]School of Public Health, Levy Mwanawasa Medical University, Lusaka, Zambia
[2]School of Public Health, University of Zambia, Lusaka, Zambia
[3]School of Medicine and Health Sciences, Eden University, Lusaka, Zambia
[4]School of Education, University of Zambia, Lusaka, Zambia

**Correspondence to**
Dr Cephas Sialubanje; csialubanje@yahoo.com

## ABSTRACT

**Objective** To explore the health professionals' and community members' perspectives on the factors influencing transmission of the novel COVID-19 in Zambia.

**Design** An exploratory qualitative study using in-depth interviews as data collection technique.

**Setting** Four primary healthcare facilities and local communities of Lusaka city and Chirundu international border town under Lusaka province, Zambia.

**Participants** Purposive sampling of 60 study participants comprising health professionals (n=15) and community members (n=45). Health staff were health inspectors and surveillance officers. Community members included public market traders, civic and religious leaders, immigration officers, bus and international truck drivers.

**Results** Both health professionals and community members were aware of the COVID-19 pandemic, the preventive and control measures. Nevertheless, stark differences were observed on the two groups' perspectives on COVID-19 and the factors influencing its transmission. Most health staff expressed high personal risk and susceptibility to the disease and a positive attitude towards the prevention and control measures. Conversely, myths and misconceptions influenced most community members' perspectives on the disease and their attitude towards the COVID-19 guidelines. Participants were unanimous on the low levels of adherence to the COVID-19 preventive and control measures in the community. Reasons for non-adherence included limited information on COVID-19, negative attitude towards COVID-19 guidelines, social movement and travel patterns, networks and interactions, living and work conditions, water and sanitation facilities, and observation of behaviours of important role models such as politicians and other community leaders. These factors were perceived to increase the risk of COVID-19 transmission.

**Conclusion** These findings highlight important factors influencing transmission of COVID-19 in Zambia. Future interventions should focus on providing information to mitigate myths and misconceptions, increasing people's risk perception to the disease, and improving attitude towards the prevention and control interventions and mitigating structural and socioeconomic barriers.

## Strengths and limitations of this study

► The qualitative study design allowed for in-depth exploration and analysis of the subject under investigation.

► Our purposive sampling of study sample comprising health workers and community members from both urban and rural sites, with varied socioeconomic backgrounds and experiences with regard to COVID-19 allowed for triangulation and increased the validity of the study.

► Selection of research assistants experienced in qualitative methods, training them in theoretical and practical aspects of our study as well as supervision of the data collection process by an experienced research team minimised bias and assured internal validity of the study.

► Conducting the study during the first wave of the COVID-19, when the disease was still new and the number of cases and deaths was still low in the country, may have introduced bias into the study as the community members' perspectives may have changed over the three successive waves of COVID-19 experienced in the country.

## BACKGROUND

SARS-CoV-2, a novel coronavirus disease which emerged in Wuhan, Hubei, China in early December 2019 has now spread around the world.[1] The outbreak was reported to the WHO country office on 31 December 2019,[2–4] which named the new coronavirus disease as COVID-19.[5–7] Following reports of increased spread and number of confirmed cases and deaths, on 30 January, WHO declared the outbreak a Public Health Emergency of International Concern.[7] With more countries affected, on 11 March, the WHO Director General described COVID-19 as a pandemic.[7] With the disease being reported among health workers and patients with no history of contact with sea food, the mode

of transmission of the virus was established as person-to-person through surfaces, droplets and aerosols produced during coughing.[8 9] As of 23 January 2022, more than 349 million cases and 5.59 million deaths have been reported globally.[10]

The first confirmed cases of COVID-19 in Zambia were reported on 18 March 2020 and involved a Zambian couple that had travelled and stayed in France for 10 days.[11] Since then, clusters of confirmed cases were reported from Lusaka city and Kafue town. Before long, cases were reported from the Copperbelt province, Nakonde and Chirundu borders with Tanzania and Zimbabwe, respectively, and eventually from all the ten provinces in the country. To date, more than 300 000 cases and 3900 deaths have been reported in the country.[11] Following reports of the initial cases, the Zambian government enacted statutory instrument number 22[12] and instituted prevention control measures including social distancing, restriction of social gatherings by closing schools, bars and casinos, restaurants and other businesses; wearing of face masks, washing hands and sanitising. Messages on COVID-19 transmission, prevention and control were disseminated through mass media. Other measures included restriction of international travel as well as closing of international airports and restricting flights to one (Kenneth Kaunda) international airport; screening and testing; contact tracing, quarantine and isolation, and case management. The restrictive measures were intended to reduce the spread of the virus through social mixing and person-to-person contact, which promote spread of infectious diseases, including COVID-19.[12]

Social distancing and restriction of social gathering measures have often been used in epidemics[13] because they encourage the general public to avoid crowded places and drastically shift social mixing patterns.[13–16] For example, in their study conducted in China, Qui and colleagues[16] showed that social mixing patterns affect the trajectory of the outbreak within the local community. Social mixing and networks, frequency of contacts in different age groups and locations–such as schools, churches, workplaces, households, bars and other social gatherings–have a direct effect on the time-dependent and basic reproduction numbers facilitating SARS-CoV-2 transmission in the community.[17]

Currently, there is limited evidence on the extent to which these interventions are adhered to in the community. Moreover, community members' perspectives and attitude towards these measures are not known. The aim of this study was to explore the health workers' and community members' perspectives on the factors influencing transmission of COVID-19 in Zambia. Information is needed to inform design and implementation of interventions focusing on reducing transmission of the virus in the country.

## METHODS
### Study design
An exploratory phenomenological qualitative study was conducted using in-depth interviews (IDIs) as a data collection technique over a period of 12 weeks, from 3 August 2020 to 30 October 2020. Use of IDIs allowed for a detailed exploration of the subject under investigation.[18 19]

### Study setting
The study was conducted in four primary healthcare facilities and their local communities: three from Lusaka city and one from Chirundu district under Lusaka province. Lusaka is the capital city of Zambia and has an estimated population of 2.9 million.[20] Chirundu is an international border town between Zambia and Zimbabwe. It is located 115 km south-east of Lusaka city and has an estimated population of 127 600.[20] Selection of the two sites was done purposively; Lusaka was included in the study because it was an epicentre of the COVID-19 during the first wave. Chirundu town was selected because it was one of the COVID-19 hot spots in the country and it is a town of entry from the bordering countries to the southern African countries of Zimbabwe, Mozambique, Swaziland, Lesotho and South Africa–at the time, South Africa had the highest prevalence of COVID-19 in the region.[21] With regard to health services, Lusaka has two tertiary, 5 second level and several primary level healthcare facilities. In addition, there are private and faith-based healthcare providers. Chirundu has one faith-based second level hospital and several primary healthcare level facilities. The Ministry of Health headquarters and the Zambia National Public Health Institute (ZNPHI)–responsible for the COVID-19 policies, guidelines and disease intelligence- are also located in Lusaka. Services provided include specialised and primary healthcare, including health promotion on COVID-19, screening, testing and contact racing, case management and vaccination.

### Participants and sampling technique
A purposive sampling technique was used to select 60 study participants. Purposive sampling allows for selection of participants with similar experiences regarding the health problem under investigation (ie, COVID-19), while, at the same time, allowing for recruitment of participants with different demographic and socioeconomic characteristics–such as place of residence, occupation and income levels. This, in turn, helps provide insight into the similarities and differences in the participants' experiences with regard to the health problem under investigation.[18 19]

Study participants comprised 15 health workers (7 nurses and 8 public health officers) and 45 community members (18 public market traders, 6 bus drivers and 6 international truck drivers, 3 civic leaders, 3 religious leaders, 3 immigration officers and 6 community members who had recovered from COVID-19). Health workers were sampled from four primary health facilities (three from Lusaka and one from Chirundu border town). Eight health professionals worked in the primary health facilities, four worked as port health staff at the Lusaka International Airport; and the remainder worked

as surveillance officers at the ZNPHI. Community members were recruited from the local communities under the four primary health facilities. International truck drivers were selected from Chirundu border; bus drivers were selected from Lusaka intercity bus station. In order to provide insight into the similarities and differences in the IDI participants' perspectives and lived experiences with COVID-19, we sought to balance the number of participants by place of residence and occupation.[18 19]

To be included into the study, participants needed to be:

► Aged above 18 years.
► Residing in the selected area for more than 3 months (except for the international truck drivers sampled from Chirundu border).
► Health workers involved in the COVID-19 programme from the selected primary health facilities, international airport and ZNPHI.
► Bus drivers from Lusaka.

Those who were aged less than 18 years and new in the area were not eligible to participate in the study.

### Data collection procedures

Four research assistants were trained in interviewing techniques over a period of 5 days. The training comprised two phases: 3 days of theory and 2 days of practical fieldwork. Efforts were made to select research assistants who were skilled and experienced in qualitative research and spoke both English and the local languages, Nyanja, Bemba or Tonga.

Interviews were conducted at the participant's preferred place, including the place of residence or office. On average, each interview lasted between 1 hour and 1.5 hours. To ensure quality in data collection, a digital voice recorder was used. Each interview was conducted by a pair of research assistants: one facilitated the interview and took notes, the other one was in charge of the digital voice recorder. Before each interview, written informed consent was obtained from each participant; those who could not read or write were asked to mark with an 'X'. To make it easy for the study participants to understand, the consent form (online supplemental material 1) was translated into the local language (Tonga and Nyanja). Before the actual interview, each respondent was asked to complete a short demographic questionnaire. Research assistants read the consent form and the questionnaire and filled it in for those who could not read.

### Data collection instruments

Data was collected using a paper- based, unstructured interviewed guide (online supplemental material 2) that was developed by the research team. The interview guide had two predetermined themes including (1) perspectives on COVID-19, (2) factors affecting transmission of COVID-19. In addition, a short questionnaire (online supplemental material 3) was prepared to collect participant demographic and socioeconomic data. To ensure internal validity, both the interview guide and

the short questionnaire went through a rigorous development process. First, the principal investigator with vast experience in qualitative research and familiar with the subject, drafted the initial versions. The two themes in the interview guide were adapted from various sources, including the funding opportunity announcement from the National Science Technology Council website, the organisation that funded the research, review of the available literature on COVID-19 and researchers' experience in qualitative research methods. Next, the documents were shared with the research team members for their comments and feedback. The documents were revised based on the research team's comments. Two independent bilingual experts translated the documents into the local languages, Tonga and Nyanga. The translated documents were pre-tested during research assistant training and revised accordingly.

### Biosafety

To ensure their safety, research team members and study participants were provided with disposable facemasks and hand sanitisers, which they were encouraged to use consistently during training, travel and data collection process. Moreover, research team members were encouraged to observe physical distancing during training and data collection.

### Data processing and analysis

Voicerecordings from the IDIs were transcribed and translated into English by two research assistants who were proficient both in English and the local language. To check for accuracy, members of the research team back–translated 10% of the transcripts into the local language and back into English. These versions were then compared for differences and similarities while listening to the original voice recording. After verification of accuracy in transcription and translation, each transcript was then thoroughly read by one research assistant while the other one listened to the corresponding voice recording. Each translated transcript was then compared with the handwritten field notes that the research assistants prepared during the IDIs. After proof-reading and making corrections, the transcripts were saved on a password-protected computer file kept by the PI for safety. The Word documents were then imported into Nvivo 11 MAC for coding and analysis, and the categories and key sub-themes were identified (online supplemental material 2). Coding was done independently by two research team members experienced in qualitative research analysis. The derived codes and categories were later compiled and compared for differences and similarities. In order to make it easy to compare differences and similarities in the respondent perspectives by group (health worker vs community members) and residential area (urban vs rural), separate analyse were done. An inductive approach was used to analyse the qualitative data in NVivo QSR V.12, and subthemes were derived by content-analysis. The inductive approach ensured that subthemes were derived

from the predetermined themes by content-analysis and grouping all similar statements made with respect to particular themes. Quantitative sociodemographic data was analysed using descriptive statistics.

## Patient and public involvement

The study design was determined by the call for research proposals on COVID-19 research under the strategic research fund. Thus, the participants and the public were not directly involved in the conceptualisation and design of the study. However, selection of the primary health facilities and study participants was done in collaboration with the provincial and district health managers. First, an inception meeting was held with the ZNPHI and Ministry of Health staff from the Lusaka Provincial Health Office and the two districts to present and discuss the design and objectives of the study. Next, prefield meetings were held with the provincial and district managers to select primary healthcare facilities and local communities to be included in the study. Recruitment of participants (health workers and community members) was done by the by the primary health facility managers and local community leaders. Finally, a report was written and shared with key stakeholders, including the funding organisation and Ministry of Health. In addition, a dissemination meeting was held in Lusaka to share the results with the Ministry of Health and community leaders in the districts where the study was conducted.

## RESULTS

In this section, we present our study findings in three parts: summary of the participants' demographic characteristics (part 1); health workers' and community members' perspectives on COVID-19 (part 2); and the factors which influence transmission of COVID-19 (part 3).

## Demographic characteristics

The majority (60%) of the participants were female, the mean age was just above 38.12 years, and they had, on average, between two and three children. Over half (65%) of the participants were married. One-third of the participants (33.3%) had secondary school education, one-fifth (21.7%) had tertiary level education and 1.67% had never attended school at all. Concerning level of monthly income, 20% had an average income of less than K500 per month. Majority of the sample (51.7%) mentioned having had travelled out of town and 5% out of the country. The countries travelled to were South Africa, Zimbabwe and Demographic Republic of Congo. The detailed demographic and socioeconomic profile of the study participants is shown in table 1.

## Theme 1: perspectives on COVID-19

In this theme, we present the findings on the health workers' and community members' perspectives on COVID-19 including awareness about the COVID-19, sources of information, myths and misconceptions about the disease and knowledge regarding preventive measures.

**Table 1** Sociodemographic characteristics (n=60)

| Variable | Mean (SD)/n (%) |
|---|---|
| Age | 38.12 (12.08) |
| Children | 2.9 (2.0) |
| No at home | 5.9 (3.35) |
| Sex | |
| Female | 36 (60) |
| Male | 24 (40) |
| Marital status | |
| Single | 12 (20) |
| Separated | 4 (6.67) |
| Married | 39 (65) |
| Widow | 5 (8.33) |
| District | |
| Lusaka | 36 (60) |
| Chirundu | 24 (40) |
| Group | |
| Health worker | 15 (25) |
| Community member | 45 (75) |
| Level of education | |
| Never attended | 1 (1.67) |
| Lower primary (1–4) | 4 (6.67) |
| Upper primary (5–7) | 12 (20) |
| Junior secondary (8–9) | 10 (16.67) |
| Senior secondary (10–12) | 20 (33.33) |
| College | 8 (13.33) |
| University | 5 (8.33) |
| Occupation | |
| Health worker | 15 (25) |
| Community member | 45 (75.0) |
| Level of income | |
| <K500 | 12 (20.0) |
| K500–K999 | 15 (25.0) |
| K1000–K1499 | 5 (8.33) |
| K1500–K1999 | 3 (5.0) |
| >K2000 | 25 (41.67) |
| Travelled out of town | |
| Yes | 31 (51.67) |
| No | 29 (48.33) |
| Travelled out of the country | |
| Yes | 3 (5.0) |
| No | 57 (95.0) |
| Countries travelled to | |
| South Africa, Zimbabwe, DRC | |

DRC, Demographic Republic of Congo.

## Awareness about COVID-19

All the 15 health workers and most (60%) community members (except some bus drivers and public market traders) from both study sites were aware of COVID-19 and explained that they had seen on television (TV) how it had spread to many countries and that people in other countries had died from the disease.

> When you watch on TV you can see the massive deaths which have happened in other countries, it shows that there is COVID (Community member, Immigration Officer, Chirundu district)

In contrast, some community members, especially public market traders and bus drivers from Lusaka mentioned that, although they had heard about people dying, they did not know whether the deaths were due to COVID-19 or not. They also explained that some people did not believe the disease existed because they had not seen it as they did with other diseases like Ebola, and that they would only believe if they saw someone dying from COVID-19. Asked whether they had heard the reports from the Ministry of Health on the number of people who had died in the country, these participants explained that, in Africa COVID-19 did not kill as many people as it did in other continents, and that if COVID-19 really existed in Zambia, most people would have died already.

> There is nothing conclusive. I need to see the corpses. Not until I see a dead body of someone and they say this one has died from Covid-19COVID-19 [Bus driver, Lusaka district].

## Sources of information about COVID-19

Both the health workers and community members who had heard about COVID-19 cited the Ministry of Health and the media (radio and TV) as the main sources of information on COVID-19. Especially the health workers, civic leaders and people who had recovered from COVID-19 explained that the information included the mode of transmission of the virus, the preventive and control measures and the need to adhere to the guidelines. By contrast, some participants, especially community members interviewed in Chirundu, explained that it was difficult to access information on COVID-19. They also expressed ignorance with regard to the source of the COVID-19 and the symptoms it causes.

> The doctor (Minister of Health) said it on TV. He said that the disease is real and that the number of people with COVID-19 in the country has increased [Community member, Lusaka district]

## Myths and misconceptions about COVID-19

Study participants expressed different views on beliefs about the existence, origin and mode of transmission of the disease. Most health workers believed that the disease was real and that it was caused by the virus spread through the air and contact with other people and objects. In contrast, community members, including public market traders and bus drivers expressed myths and misconceptions about the COVID-19. They believed that someone had created the disease in order to eliminate the African population, and that foreigners (Chinese) who were interested in opening mines in African countries, like Congo, brought the disease so that local people could die, and that, eventually, the foreigners would take over the mines. Further, some community members including religious leaders believed that the disease was a mark of the beast recorded in the Bible (Revelation 13). They explained that most people did not believe COVID-19 was real, and that government was playing politics and that politicians were working with foreigners to gain their support and make money for themselves. Some attributed COVID-19 to radiation from cell phones and industries, and that most people died because of radiation from 5G Networks. Consequently, they wondered why they were being asked to test for the disease and to wear facemasks.

> I am completely confused, is it really a natural disease? I think it is a fabricated disease or something like that. It all came from the Chinse people, and that other people are just inheriting it (Public market trader, Chirundu district).

## Knowledge about COVID-19 preventive measures

Health workers and community members expressed contrasting perspectives on their knowledge about preventive and control measures. Most health workers and community members from Lusaka expressed knowledge about the COVID-19 prevention and control guidelines. They explained that they helped disseminate the information in the communities where they worked from and encouraged people to adhere to the guidelines including wearing facemasks, social distancing and avoiding overcrowded places, ensuring hand washing and sanitising. Conversely, most community members including those from Chirundu border, expressed ignorance about the preventive and control measures. They explained that it was difficult to get information about COVID-19 because of the limited access to TV, radio, internet and mobile phone network. Especially bus drivers and public market traders mentioned that they only hear from people in the community and that most information was incorrect.

> They tell us to wear face masks and wash our hands with soap, but most of us do not have anyone to explain to us (Public Market Trader, Lusaka city)

### Theme 2: factors influencing transmission of COVID-19

In this section, we present the findings on the factors that influence transmission of COVID-19 in the community. They include living and working conditions, travel and social movement patterns, and social networks.

### Living and working conditions

Both health workers and community members (ie, those who believed in the existence of the disease) were

unanimous on living and working conditions being important factors driving the COVID-19 transmission. Overcrowding in some homes and work places made it difficult to maintain social distance.

Public health officers explained that staff shared the office space and majority did not have enough hand-washing or sanitising facilities; they only used one shared sanitiser and the hand washing facility. They added that some offices and homes did not have running water; where it was present, the supply was erratic and made it difficult for people to follow the guidelines on hand-washing and other hygienic purposes.

> You know our tradition, parents use their own bed-room, then the kids, maybe there are 5 girls, and they use one bedroom. If there are 6 boys they will use one bedroom [Immigration officer, Chirundu district].

### Attitude towards preventive measures

Participants expressed different attitudes towards the preventive measures. Health workers and some community members (international truck drivers and those who had recovered from or seen someone with COVID-19) expresseda positive attitude towards preventive measures. They explained that the measures were beneficial to the population. The main benefits were protection against the virus, helping people maintain hygienic standards in the community. They explained that, in the beginning, community members never believed in the reality of the disease until they noticed that even in Zambia, people started dying; some were being quarantined. Health workers explained that there was a need for people to follow the prevention and control guidelines from the Ministry of Health, especially during the cold season, when the risk of transmission of the virus was high.

> What I think is that COVID is there, so the most im-portant thing is to just follow what they say we should do [Public Health Officer, Lusaka district]

In contrast, most community members expressed a negative attitude towards the COVID-19 preventive meas-ures. They wondered why they were being asked to wear facemasks or avoid mixing with their friends or attending important social functions like church services, weddings or drinking parties. Some participants also explained that it was not clear why even children were told to wash their hands or wear facemasks.

> To be honest people do not like these things, they do not follow because they say, it is a lie, Corona is not there. They are saying because there is no Corona so they do not wear the mask [Community member, Lusaka district].

### Adherence to preventive measures

There was consensus among the participants concerning non-adherence to preventive measures in the commu-nity–which was seen as one of the important factors contributing to the spread of COVID-19 in the popula-tion. Health workers observed that, although some people followed the guidelines–wore facemasks and used hand-sanitisers– majority did not. They had continued mixing and interacting as before, especially in the markets and buses. Asked why they did not adhere to the guidelines, most community members explained that people in their communities believed that preventive measures came from other countries and local people, especially Africans, were just being told to follow. They also explained that most people were discouraged by what they saw among political leaders who held meetings and political rallies without following the guidelines. They explained that politicians 'played politics' as they did not maintain social distance or wear facemasks when holding pre-election political rallies. Similarly, most bus drivers confirmed that most passengers did not observe the guidelines when trav-elling on public transport.

Participants had consensus on the importance of social gatherings such as weddings, kitchen parties, church services, taverns and funerals as virus super spreader events. Nevertheless, they observed that nothing had changed with regard to social mixing patterns; people were used to their old way of life of greeting and hugging their friends when they met. They argued that people did not feel good to stand a metre or an arm apart because of the new rules. Similarly, participants, especially health workers, civic and religious leaders, confirmed that most families had continued visiting and inviting friends or extended family members to their homes for family and social functions such as parties and beer drinking. Although some people did not attend funerals and only provided monetary and other material support, many community members did. Further, shops and churches remained open and people continued going to the market, shops for their businesses, shopping and attending church services.

> The only time we can say social distance, is if we come to the market, in the shops, or church. There is no social distance when you go to the political rallies [Religious leader, Lusaka district]

### Travel and movement patterns

Both health workers and most community members (except some bus drivers) confirmed that travel within and outside the country was a major factor influencing transmission of the virus. They observed that, despite the COVID-19 situation, local and international travel remained the same. Public transport on buses and mini-buses was the the most common mode of transport. Those who owned cars explained that they travelled in their personal cars. Reasons cited for travel were work, business, shopping, family matters (such as funerals), and other purposes.

Bus drivers explained that although they allowed fewer passengers than before, most minibuses still loaded to full capacity–this made it difficult to maintain social distance

on buses. Moreover, most passengers did not use face-masks. By contrast, some participants, especially community members and bus drivers, argued that travel did not put them at risk of infection because, before boarding on buses, passengers were made to wear facemasks and that those who did not have were not allowed to board.

> I have seen that there is no social distance on those buses. But when driving yourself yeah, there is protection because you know you are alone or maybe just the two of you, you mask up [Immigration Officer, Chirundu border].

International travel was perceived as an important driver of the COVID-19 transmission. Although some countries had closed their borders, participants observed that travel between countries, especially by road, continued. They explained that, international truck drivers spent many days away from home, in foreign countries and on transit… One driver explained that, within a space of 2 weeks, he had been to many countries including South Africa, Zimba, Zambia and DR Congo. Although they sanitise during travel, interaction with a lot of people and handling papers during transit and at the borders, put drivers and other travellers at an increased risk of getting infected or spreading the infection to others

> I am always moving to Zambia, Congo, South Africa, just like that. I was in Congo and then I connected to South Africa, that same month. I spent somewhere like 2 weeks in South Africa, then I came back [International truck driver, Chirundu border]

### Social network

Social networks were also cited as an important factor contributing to the spread of COVID-19 infection in the community. Although the frequency had reduced, people still visited their friends, workmates and relatives, and their friends. They explained that, especially, workmates visited in the evening when they knocked off and that in some instances they visited twice or thrice in a week.

> If I leave my place I go to her place and if you are to find me, mainly I am there. If I don't go to her place, she will come here, we chat and chat [Public market trader, Lusaka district].

Conversely, some participants explained that they no longer went out to interact with their friends as much as they used to before the COVID-19 pandemic. They mentioned that they had stopped attending church services and stayed at home to avoid contracting and spreading the virus. In addition, some participants explained that, although they visited family members and went to specific places for their businesses, they avoided physical meetings and gatherings; they only communicated on the phone, and that if people went home, they would only greet and not spend time together.

> From the time the issues of corona started I have never attended a wedding ceremony or funeral issues, I have never attended [Public market trader, Lusaka district].

## DISCUSSION

The aim of this study was to explore the health workers' and community members' perspectives on the factors influencing transmission of COVID-19 in Zambia. Overall, our findings suggest that most people were aware of the COVID-19 pandemic and the preventive measures. Nevertheless, several factors including myths and misconceptions and a low risk perception about COVID-19, a negative attitude towards the preventive and control measures, movement and travel patterns, working and living conditions, social gatherings, networks and interactions affected adherence to the preventive and control measures and increased the risk of COVID-19 transmission in the community.

Our findings suggest that, people were generally aware of the COVID-19 disease and the associated preventive and control measures. The Ministry of Health, media (TV, radio, internet and social media) and personal experience seem to have been the important sources of information about COVID-19 and preventive measures. This finding corroborates a previous study conducted in Egypt and Nigeria[22] which showed that, despite not adhering to preventive measures, most people have satisfactory knowledge about COVID-19 and the preventive and control measures. Nevertheless, our findings show that limited access to information on COVID-19 appear to be a major challenge, especially in rural communities where TV, radio and mobile phone reception and internet signal are poor. Consequently, a lack of information on COVID-19 seems to have led to widespread myths and misconceptions about the new disease—its existence, origin and severity—and ultimately influenced people's attitude towards prevention and control measures. Further, people's attitude towards the COVID-19 preventive measures seem to affect adherence to preventive and control measures. Participants expressing such strongly held beliefs doubted the existence of the virus and the COVID-19 disease; they expressed a negative attitude towards the preventive measures. They also expressed a low risk perception towards the severity of and their personal susceptibility to the disease. These findings highlight the importance of attitude and risk perception in influencing people's decision to adopt healthy behaviours—such as COVID-19 preventive measures.[23 24] Our findings suggest that, non-adherence to preventive measures was a major contributing factor to the spread of the virus in the community. Interventions should focusing on preventing and mitigating myths and misconceptions about COVID-19 by providing accessible correct, consistent and timely information on COVID-19 in rural areas—through improved TV, radio reception and internet and mobile phone connectivity. Interventions should also focus on provision of information about the benefits of

adherence to prevention and control measures in order to increase community members' attitude and personalised risk perception. Interventions focusing on improving personalised risk perception, behavioural beliefs and attitude have been shown to increase adoption of and adherence to a target health behaviour.[24]

Life style, living and work conditions were identified as important drivers of the COVID-19 transmission. For example, the family size and living conditions in most homes—especially in high density urban neighbourhoods and rural areas where a number of family members shared sleeping spaces—made it difficult to practice social distancing in the home. Similarly, our findings suggest that sharing and overcrowded offices, poor water supply and inadequate sanitisers and washing facilities in places of work posed an important risk for the transmission of the virus in the work setup. Moreover, poor water supply and overcrowding, made it difficult to observe social distance and adherence to hand washing and hygienic practices, especially in public markets. These findings corroborate previous studies.[25–27] For example, a systematic review of 68 studies conducted by Zar and colleagues[26] found that poor living conditions in low-income and middle-income countries, including lack of adequate sanitary facilities, running water and overcrowding pose challenges to adherence to prevention and control guidelines among urban slum residents and facilitate transmission of SARS-CoV-2. Further, Ahmad and others[27] reported a higher risk of infection and death from COVID-19 in counties with poor housing compared with those with modern housing facilities. Policies and intervention on COVID-19 prevention would benefit from improved housing, work condition and sanitation and water supply.

Moreover, local and international travel patterns were found to be a major risk factor for community transmission and importation of cases into the country, respectively. Although a number of people used private transport for local travel when going for business, work, family matters and other purposes, majority travelled on public transport such as buses and minibuses where adherence to social distance is difficult. Although international travel by the general public had reduced at the time of the study, travel by road was continued as before. Interaction with their fellow drivers and other travellers as well as customs and immigration officers all put international truck drivers at an increased risk of contracting and transmitting the disease. These findings corroborate previous studies[17 28] which highlighted the importance of international and local travel in influencing spread of the virus. For example, in their study[28] conducted in the USA, Davis *et al* reported international travel as the key driver of the introduction of SARS-CoV-2 in the West and East Coast metropolitan areas that could have been seeded as early as late-December, 2019. The study also showed that for most of the continental states the largest contribution of imported infections arrived through domestic travel flows. Similarly, a study conducted in

China[17] showed that although travel to foreign countries was responsible for the importation of initial infections into the country, social mixing patterns affected the trajectory of the outbreak within the local community. Our findings highlight the benefits to be gained from interventions focusing on limiting social movement through restriction of local and international travel.

Finally, our findings suggest that social networks, frequency and duration of contacts are an important risk factor for the spread of the virus. Despite the government guidelines on restricting social gatherings, most people maintained their social networks through social functions such as wedding ceremonies, church services, kitchen parties, funerals, casinos and bars, and home visitations. This finding corroborate previous studies[17 29] which reported the importance of social gatherings in facilitating spread of the virus and increasing the basic reproductive number. For example, a study by Alimohamadi *et al* conducted during the early phase of the pandemic highlighted the effect of social interaction and networks in different age groups and locations on the time-dependent and basic reproduction numbers governing SARS-CoV-2 transmission in the community.[29] The study concluded that reducing the number of contacts within the population is a necessary step to control the epidemic. Thus, policies regulating social interactions could help reduce transmission of the virus.

Limitations of our study should be noted. First, like all qualitative studies, our findings are based on the few IDI participants' views. We could not conduct focus group discussions to compare and confirm the findings due to safety considerations and logistical challenges. Further, the study was conducted during the first wave of the COVID-19 pandemic when the disease was still new and the number of cases and deaths was still low in the country. Zambia has experienced four successive waves. It is not clear whether community members' and health workers' perspectives have changed over these successive waves of COVID-19 in the country. However, the selected participants' perspectives may not have been representative of the views of the other members of the community, especially those who had tested positive to the virus and experienced it. Although Chirundu and Lusaka districts had one of the highest number of COVID-19 cases, most study participants had not experienced the disease. The reason for this selection was to have participants who could provide insightful information on the disease, and not necessarily having experienced it.

Despite these limitations, our study has several strengths. First, the study provides the first in-depth assessment and analysis of the health workers' and community members' perspectives on the factors influencing transmission of the virus in the two districts that were hotspots for COVID-19 during the first wave in Zambia. Second, triangulation through our purposive selection of the study participants comprising health workers and community members from both urban and rural sites, with varied socioeconomic backgrounds and experiences with regard

to COVID-19, as well as the use of trained research assistants and rigours data analysis conducted by a trained and experienced team minimised bias and increased the internal validity of the study findings. Finally, our findings provide important insights into the factors influencing transmission of COVID-19 in Zambia can serve as important targets for policy and intervention design for prevention of COVID-19 transmission in Zambia and other countries with similar demographic and epidemiological contexts. As far as we know, this is the first qualitative study conducted on this subject in Zambia.

## CONCLUSION

These findings highlight important factors influencing transmission of the coronavirus in Zambia including limited information about COVID-19, myths and misconceptions about the disease, negative attitude towards and non-adherence to prevention and control measures, negative behaviour of the role models such as politicians and other important community leaders, living and working conditions, travel and movement patterns, social gatherings, networks and interactions. Future prevention interventions should focus on: (1) increasing access to information about COVID-19; (2) increasing people's risk perception and personal susceptibility to the disease, (3) improving attitude towards the benefits of preventive and control interventions, (4) limiting local and international travel; (5) restricting social gatherings and (6) mitigating structural and socioeconomic barriers–by improving housing and work conditions, provision of safe water and sanitary facilities—in order to increase adoption of and adherence to preventive measures.

Finally, further research, in form of a survey with a quantitative design, is needed to measure people's attitude towards prevention measures and determine the levels of adherence to the life-saving interventions. Research with a longitudinal design is also required to test whether adherence to the preventive and control measures actually leads to reduced transmission of the Corona virus and improved health outcomes. Findings from such studies can serve as basis for design of public health policy on preventive and control interventions as well as advocacy for adherence to these measures. Currently, this evidence is non-existent.

**Acknowledgements** We thank Eden University administration, Dr Flavien N. Bumbangi and Mr Kennedy Chishimba for the logistical support they provided during the training of data collectors and data collection process. Our gratitude also goes to the study participants for their valuable time and input into the study.

**Contributors** All authors designed the study. Under the oversight of CS, NM and LL, HKK, MN, supervised the data collection process. CS and DCS conducted data analysis. CS wrote the first draft of the manuscript. PIS, NM and DCS revised the manuscript. FMG advised on the final manuscript. All authors read, commented on and approved the final manuscript. CS had access to the data, controlled the decision to publish and is the study guarantor.

**Funding** This work was supported by the National Science and Technology Council (grant number NSTC 101/6/8), as part of the Emergency COVID-19 research grant under the strategic research fund.

**Competing interests** None declared.

**Patient and public involvement** Patients and/or the public were involved in the design, or conduct, or reporting, or dissemination plans of this research. Refer to the Methods section for further details.

**Patient consent for publication** Not applicable.

**Ethics approval** Ethical approval was obtained from the University of Zambia Biomedical Research Ethics Committee (UNZA BREC, ref 1132–2020); permission to conduct the study was obtained from the National Health Research Authority.

**Provenance and peer review** Not commissioned; externally peer reviewed.

**Data availability statement** Data are available on reasonable request. Data are available on reasonable request to the corresponding author and with permission of the UNZABREC Institutional review board.

**ORCID iD**
Cephas Sialubanje http://orcid.org/0000-0002-9077-1436

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
