## [Reviewer comments · BMJ Open]

ARTICLE DETAILS

TITLE (PROVISIONAL)	Perspectives on Factors Influencing Transmission of Coronavirus disease 19 (COVID-19) in Zambia: A qualitative study of Health Workers and Community Members
AUTHORS	Sialubanje, Cephas; Sitali, Doreen; Mukumbuta, Nawa; Liyali, Libonda; Sumbwa, Phyllis; Kamboyi, Harvey; Ng'andu, Mary; Goma, Fastone

VERSION 1 – REVIEW

REVIEWER	Arnout, B King Khalid University
REVIEW RETURNED	17-Oct-2021

GENERAL COMMENTS	Thank you for sharing your study. Please write which qualitative research design was applied (phenomenology, grounded theory, etc). And also add limitations and future directions, and also it is necessary to add the tables about the codes resulted from the data collected analysis. With my best wishes
--

REVIEWER	Mashreky, Saidur Centre for Injury Prevention and Research Bangladesh
REVIEW RETURNED	25-Nov-2021

GENERAL COMMENTS	Appreciate authors for conducting interesting study on COVID 19 pandemic. Study title and object is clearly written. Study design is selected correctly according to the research question and study objective. Methods of the study described adequately. Result and discussions are well presented. Conclusion was made according to the study findings. Page number 15-line number 60 there may be duplication of words
--

REVIEWER	Padhi, Bijaya Post Graduate Institute of Medical Education and Research, Community Medicine and School of Public Health
REVIEW RETURNED	27-Nov-2021

GENERAL COMMENTS	The study explored the health professionals' and community members' perspectives on the factors influencing transmission of the novel coronavirus disease 2019 (COVID-19) in Zambia. The authors used in-depth interviews to collect data from health professionals (n=15) and community members (n=45). Authors
--

	reported that “study participants expressed a negative attitude towards the preventive and control measures. Both groups of participants unanimous on the low levels of adherence to the COVID-19 preventive and control measures in the communities”. The findings are a concern in local/global measures for preventive approaches and need further investigation. I commend the authors for conceptualizing this study, as the results certainly add value to the literature. The study used a valid research question and hypothesis, with a relevant theory to which the research question is being posed. The study also applies correct and transparent methodology, and the study design and materials are clearly laid out. The language and presentation are clear and adequate, figures and tables are in line with scientific norms and standards. This is a well-written article; however, some areas need improvements:  1) The developmental steps used in study tools should be described in the method section. 2) Psychometrics of the survey instruments should be provided. 3) Did the authors use any triangulation approach for data quality control? 4) In discussion, the authors stated that “The aim of this study was to explore the psychosocial and cultural factors influencing transmission of COVID-19 in Zambia in order to inform design and implementation of interventions focusing on reducing transmission of the virus in the country.” I believe the authors should be confined to the primary objective of the study. 5) Some typo on page#15 “Interestingly, our findings suggest that people’s attitudes towards the COVID-19 preventive measures affects their adherence to adherence to preventive measures– such as consistent and correct wearing”. Repeat use of “adherence to.”
--	--

VERSION 1 – AUTHOR RESPONSE

Reviewer: 1

1. The reviewer commended the authors for sharing our study and asked us to write which qualitative research design was applied (phenomenology, grounded theory, etc).

Response: We thank the reviewer for this comment. We have now added the word “phenomenological” in the methods section (page 5 line 12)

2. The reviewer recommended that we add limitations and future directions,

Response: We appreciate the reviewer’s comment. The limitations are there at the end of the discussion section in the document. We have also edited this section to improve it (19 lines 21- 34). We have also added and discussed a comprehensive list of recommendations in the discussion and conclusion sections.

3. The reviewer asked us to add the tables about the codes that resulted from the data collected for analysis.

Response: We have added the codebook/tables as supplementary material 2 (page 8 line 28).

Reviewer: 2

1. The reviewer appreciated us for conducting an interesting study on COVID 19 pandemic, that the study title and object were clearly written, study design was selected correctly according to the

research question and study objective, methods of the study were described adequately, result and discussions were well presented, and that the conclusion was made according to the study findings.

2. The reviewer observed that there might be a duplication of words on page number 15-line number 60.

Response: We appreciate this observation by the reviewer; we have now corrected this repetition. We have also edited most of the discussion to improve the quality of the language and flow of ideas

Reviewer: 3

The reviewer commended the authors for conceptualizing the study, and that the results certainly add value to the literature.

1. The reviewer recommended that the developmental steps used in study tools should be described in the method section.

Response: We thank the reviewer for this guidance. We have now provided more detail on the development process for the data collection tool (page 7 lines 29-34 and page 8 lines 1-4)

2. The reviewer recommended that the Psychometrics of the survey instruments should be provided.

Response: Since the study is qualitative in design, we did not use a survey instrument. Rather, we developed and used an interview guide, supplementary material 1 (see response 1 above, page 7 lines 29-34 and page 8 lines 1-4). As such, it was not practical to measure its validity and reliability (Psychometrics). However, due diligence was paid during its development process; the instrument was pre-tested during the training of data collectors, and revised accordingly.

3. The reviewer wanted to know whether we used any triangulation approach for data quality control.

Response: We appreciate the author for raising this concern. We used triangulation through the selection of study sites and data sources (study participants), coding and data analysis as described below and believe that these triangulation approach helped improve our data quality: a) selection of study sites: We collected data from both rural and urban sites and compared the findings (page 5 lines 30-34); b)selection of study participants: We interviewed health workers and community members and compared and contrasted the findings (page 6 lines 20-34); c)coding and data analysis: Coding was done independently by two research team members experienced in qualitative research analysis. The derived codes and categories were later compiled and compared for differences and similarities. Separate analyses were done by group (health worker versus community members) and residential area (urban versus rural) in order to make it easy to compare differences and similarities in the respondent perspectives (page 8 lines 17-34)

4. The reviewer was concerned that in the discussion, the authors should be confined to the primary objective of the study.

Response: We appreciate this comment by the reviewer. We have revised the first paragraph of the discussion to reflect the primary objective of the study (page 16 lines 25-26)

5. The reviewer observed some typo on page#15 and repeat use of “adherence to.”

Response: We have edited this repetition and corrected all the typo errors in the document (see our response under reviewer 2 response 2 above)

Editor’s comments

1. The editor advised that we revise our title so that it includes our study design (qualitative study), that this is the preferred format for the journal

Response: We have revised the title. It now reads as: “Perspectives on Factors Influencing Transmission of Coronavirus 19 (COVID-19) in Zambia: A qualitative study of Health Workers and Community Members

2. The editor requested us to carefully check the paper for typographical/ grammatical errors e.g. “Both groups of participants unanimous on the low levels of adherence to the COVID-19 preventive and control measures in the community” (abstract)

Response: We have proof-read the whole document and corrected all the typo and grammatical errors.

3. The editor advised us to include the interview guide as a regular supplementary file and refer to this in the methods section

Response: We appreciate this comment; in our first submission, we included the interview guide as an appendix. We have now uploaded it as a regular supplementary file 1. We have also referred to it in the methods section of the manuscript (page 7 line 24).

4. The editor advised that, along with our revised manuscript, we provide a completed copy of the SRQR checklist.

Response: We have downloaded the SRQR checklist and completed it.